# Multi-Agent Task Allocation with Multiple Depots Using Graph Attention Pointer Network

Wen Shi [1,*] and Chengpu Yu [1,2]

1 School of Automation, Beijing Institute of Technology, Beijing 100081, China; yuchengpu@bit.edu.cn
2 Beijing Institute of Technology Chongqing Innovation Center, Chongqing 401120, China
* Correspondence: 3220200783@bit.edu.cn

**Abstract:** The study of the multi-agent task allocation problem with multiple depots is crucial for investigating multi-agent collaboration. Although many traditional heuristic algorithms can be adopted to handle the concerned task allocation problem, they are not able to efficiently obtain optimal or suboptimal solutions. To this end, a graph attention pointer network is built in this paper to deal with the multi-agent task allocation problem. Specifically, the multi-head attention mechanism is employed for the feature extraction of nodes, and a pointer network with parallel two-way selection and parallel output is introduced to further improve the performance of multi-agent cooperation and the efficiency of task allocation. Experimental results are provided to show that the presented graph attention pointer network outperforms the traditional heuristic algorithms.

**Keywords:** task allocation; attention mechanism; multi-agent system



## 1. Introduction

Artificial intelligence (AI) is one of the greatest technologies in the 21st century. In recent years, with the rapid development of the new generation of AI technology, the research and application of intelligent unmanned systems have received widespread attention. Intelligent unmanned systems can not only complete simple and repetitive tasks but also complex tasks that are difficult for humans. The core advantage of intelligent unmanned systems lies in achieving high performance while having extremely low costs. Multi-agent collaboration is one of the most important technologies in intelligent unmanned systems; it is more effective than single-agent technology when facing complex task requirements. Multi-agent collaboration technology describes the coordination of multiple agents to achieve stronger performance, better efficiency, and higher adaptability than simple addition. The application of multi-agent systems will be more and more extensive with the progress of AI technology, which will have a profound influence on future daily life and industrial production.

Task allocation is one of the most important problems in multi-agent collaboration; it has been a very important research topic in recent years. Task allocation mainly solves the problem of allocating tasks to intelligent agents with reasonable strategies in complex environments, so that the performance and efficiency of task completion are as high as possible. The specific goal of task allocation is to ensure the constraint conditions and to maximize the optimization of completion time and resource consumption. At present, the technologies of multi-agent task allocation are not good enough, so improving and innovating methods and algorithms is crucial.

The task allocation problem is usually represented as a combinatorial optimization problem (COP). The traveling salesman problem (TSP) is a common single-agent task allocation problem which is classical in the combinatorial optimization area. The TSP describes how a travel merchant (agent) can visit multiple cities and return to the original city of departure with the minimum path. It only needs to consider the order of task

execution of a single agent. Multi-agent task allocation problems are more complex as they need to consider not only the execution order but also the executor of each task. Multiple traveling salesman problems (MTSPs) [1] and vehicle routing problems (VRPs) [2] are well-known for multi-agent task allocation. The MTSP expands the TSP to the multi-agent scenario. Multiple agents start from the same starting point and finally return to this point after completing a series of tasks. The goal of this problem is to obtain a set of task execution plans that have the shortest total path. VRP is a kind of problem that adds constraints on the basis of MTSP, such as the capacitated vehicle routing problem (CVRP) with capacity constraints and the split delivery vehicle routing problem (SDVRP) with split constraints [3]. It is noteworthy that these kind of multi-agent task allocation problems require that all agents must start from and return to the same point. The complexity and difficulty of solving the multi-agent task allocation problem are far greater than the single-agent task allocation problem. In recent years, many scholars have focused on this kind of multi-agent task allocation problem, and have developed a number of solutions [4].

For the concerned multi-agent task allocation problem with multiple depots, the initial positions of all agents need to be taken into account. This kind of problem can be occasionally confronted in real life, and has important applications in many fields, such as multiple unmanned aerial vehicles (UAVs), multiple unmanned ground vehicles (UGVs), and multiple unmanned surface vessels (USVs). When the environment is dynamic and changes during task execution, the subsequent allocation of agents can be regarded as a new task allocation problem with multiple depots. Similarly, in the process of multi-agent collaboration, if there is an emergency of some agent failures, a new solution can be obtained immediately to handle the emergency. However, the multi-agent task allocation problem with multiple depots is more difficult than the single-agent or the single-depot problem due to the diverse locations of the agents. So far, there are few satisfactory solutions in terms of accuracy and efficiency.

For the NP-hard task allocation problem, there are two types of classical algorithms: optimization algorithms [5] and heuristic algorithms [6]. The optimization algorithm can yield an optimal solution, but the computational burden will increase exponentially, often becoming too great to be accepted. Typical optimization algorithms for task allocation problems are simplex algorithm, branch and bound, etc. In contrast, heuristic algorithms are more practically useful. Although a heuristic algorithm can only yield an approximate solution, its computational efficiency is much higher than the optimization method. Usually, heuristic algorithms can be modified by changing the associated basis so as to achieve better results or higher efficiency. However, heuristic algorithms usually require formulating rules which may be highly dependent on human expertise and experience. The commonly used heuristic algorithms in task allocation include the genetic algorithm (GA) [6], ant colony optimization (ACO) [7], particle swarm optimization (PSO) [8], immune algorithm (IA) [9], tabu search (TS) [10], simulated annealing (SA) [11], etc.

After heuristic algorithms had been widely used for decades, deep learning methods for task allocation began to rapidly develop [12]. The encoding method, based on the attention mechanism [13], and the decoding method, based on the pointer network [14], are two important parts of the deep learning method of task allocation.

In the rapid development process of deep learning, the sequence generative model is one of the most representative and most developed models [15]. It is now relatively mature in machine translation [16] and speech recognition [17]. The attention mechanism and the transformer model [18] are the most important achievements in sequence generation. In traditional recurrent neural network (RNN) [19], it is difficult to summarize all the context details in the sequence encoding process, and attention mechanisms can select more important information from a large number of contexts. The attention mechanism determines the correlation of context details that do not rely on time sequence in order to extract more valuable detail segments from the input sequence. Specifically, the attention mechanism calculates a variable context vector by weighted averaging and then adds the context vector as additional information into the RNN to output the final sequence. The

transformer model [18] is a new attention mechanism proposed in 2017. The transformer attention mechanism no longer relies on the traditional RNN model, but only uses attention mechanisms to build the encoder and decoder. A self-attention mechanism was proposed in the transformer model, which completes parallel encoding of all input segments to greatly reduce training time. In addition, a multi-head attention mechanism was proposed to extract different features through multiple independent single-head attention layers, further improving the performance of the model.

The pointer network (Ptr-Net) [14] is an important application of the attention mechanism in task allocation problems which solved the TSP using a deep learning method. Ptr-Net combines a sequence generative model and the graph attention network (GAT) [20]. The encoder of Ptr-Net completes feature extraction of task points using the attention mechanism to obtain graph embeddings, and the decoder outputs the generation of task sequences. The output of Ptr-Net is a pointer to an input element, which is then sorted in a new order. Specifically, Ptr-Net calculates the state vector of the current moment by the pointer and the state vector of the previous moment, then uses the attention mechanism to calculate the weight of all input elements corresponding to the current state, and finally outputs the pointer according to the weight value. In Ptr-Net, all elements of the input sequence are calculated independently of each other, so changing the order of the input sequence will not change the final result. Ptr-Net contributes enormously in the task allocation area, which greatly improves the computational speed while ensuring task completion accuracy.

On the basis of Ptr-Net, the reinforcement learning method [21] was proposed to overcome the difficulty of label collection in COPs. The attention model [4] extended the pointer network to multiple agents, effectively solving several VRP problems. The relational attention model [22] modified the attention model and improved the performance and computational efficiency. However, the above mentioned methods cannot handle complex multi-agent COPs, and only a few studies have explored the multi-agent task allocation problem with multiple depots [23,24].

Based on the above literature review, a multi-agent task allocation method with multiple depots is proposed, where an improved encoder–decoder architecture is developed. A full-graph encoder with multi-head attention mechanism and a two-way selection decoder based on a pointer network are built. The encoder is an improved graph attention network, which uses the multi-head attention mechanism to calculate the features of all agents and tasks. The decoder is an improved pointer network which outputs the orders of task execution in parallel by a two-way selected model. The proposed method can yield several sequences which correspond to the task execution orders of multiple agents. This encoder–decoder architecture is trained and tested using the reinforcement learning method with baseline, which can calculate the selection probability of each agent and each task, and output the final routes of multiple agents in parallel.

The main contributions are as follows.

- A full-graph encoder is built to enhance the correlation between agents and tasks by calculating the features of the agents, tasks, and the current state.
- A two-way selection network of agents and tasks is proposed to ensure the performance of task allocation.
- A parallel pointer network is designed to output all the routes at the same time, which enables effective cooperation among multiple agents.
- The experimental results show that the proposed task allocation algorithm outperforms traditional heuristic algorithms in terms of effectiveness and efficiency.

This paper is organized as follows. Section 2 formulates the multi-agent task allocation problem with multiple depots. Section 3 builds an end-to-end model with a full-graph encoder and a two-way selection decoder. Section 4 shows the performance of our proposed model in the multi-agent task allocation problem with multiple depots. Section 5 provides conclusions and future work.

## 2. Problem Description

For the concerned multi-agent task allocation problem with multiple depots, there are multiple agents and multiple tasks. Each agent is able to execute multiple tasks, and each task only needs one agent to execute once. All agents need to start from their respective depots and cooperate to complete all tasks. The goal of this problem is to obtain a set of task execution plans with the shortest total path.

For a problem setting $s$, there are $m$ agents $\{A_1, \ldots, A_m\}$ and $n$ tasks $\{T_1, \ldots, T_n\}$. The depot coordinates of the agents and the task coordinates are represented by $\{\mathcal{C}_{A_1}, \ldots, \mathcal{C}_{A_m}\}$ and $\{\mathcal{C}_{T_1}, \ldots, \mathcal{C}_{T_n}\}$, respectively. $\mathcal{C}_A$ and $\mathcal{C}_T$ are the position vectors in 2D or 3D space.

The execution plan of the problem setting $s$ can be represented as:

$$\pi(s) = \{\pi_1(s), \ldots, \pi_m(s)\} \tag{1}$$

$$\pi_j(s) = (T_{j1}, \ldots, T_{jk_j}), j \in \{1, \ldots, m\} \tag{2}$$

where $\pi(s)$ is the set of all $m$ agents' task execution sequences and the $\pi_j(s)$ is the task execution sequence of the $j$-th agent. For the sequence $\pi_j(s)$, $T_{j1}$ represents the first task in the sequence and $T_{jk_j}$ represents the last task, and $k_j$ is the task sequence length of the $j$-th agent.

The multi-agent task allocation problem is formulated as:

$$\text{Minimize} \sum_{j=1}^{m} \mathcal{L}(\pi_j(s)), \tag{3}$$

subject to

$$\mathcal{L}(\pi_j(s)) = \|\mathcal{C}_{A_j} - \mathcal{C}_{T_{j1}}\|_2 + \sum_{h=1}^{k_j-1} \|\mathcal{C}_{T_{jh}} - \mathcal{C}_{T_{j(h+1)}}\|_2. \tag{4}$$

$$\cup_{j=1}^{m} \pi_j(s) = \{T_1, \ldots, T_n\}, \tag{5}$$

$$\pi_j(s) \cap \pi_l(s) = \varnothing, \quad \forall j, l = 1, \ldots, m, \tag{6}$$

Equation (3) is the objective function to minimize the sum of the path lengths of all routes. Equation (4) provides an expression for the path length of the route $\pi_j(s)$. Finally, Equations (5) and (6) are the constraints that all tasks need to be completed, and each task must be completed by only one agent.

## 3. Methods

A graph attention pointer network is proposed for the multi-agent task allocation problem with multiple depots. Several improvements are made in both the encoder and decoder based on Kool's encoder–decoder framework [4], so that the model can effectively handle the multi-agent task allocation problem with multiple depots.

The proposed model consists of a full-graph encoder and a two-way selection decoder. First, the information of all depots and tasks are sent to the graph attention pointer network to seek the relationship of all points in the graph. Second, the two-way selection decoder chooses the bidirectional matching between the agent and the task, and outputs multiple sequences at the same time. Finally, the model is trained by the policy gradient deep reinforcement learning algorithm with baseline.

The deep learning model in this paper transforms the multi-agent task allocation problem into a node ordering problem in a graph structure. The encoder constructs the graph structure and the decoder obtains node sequencing. The details of the proposed model are shown in Figure 1.

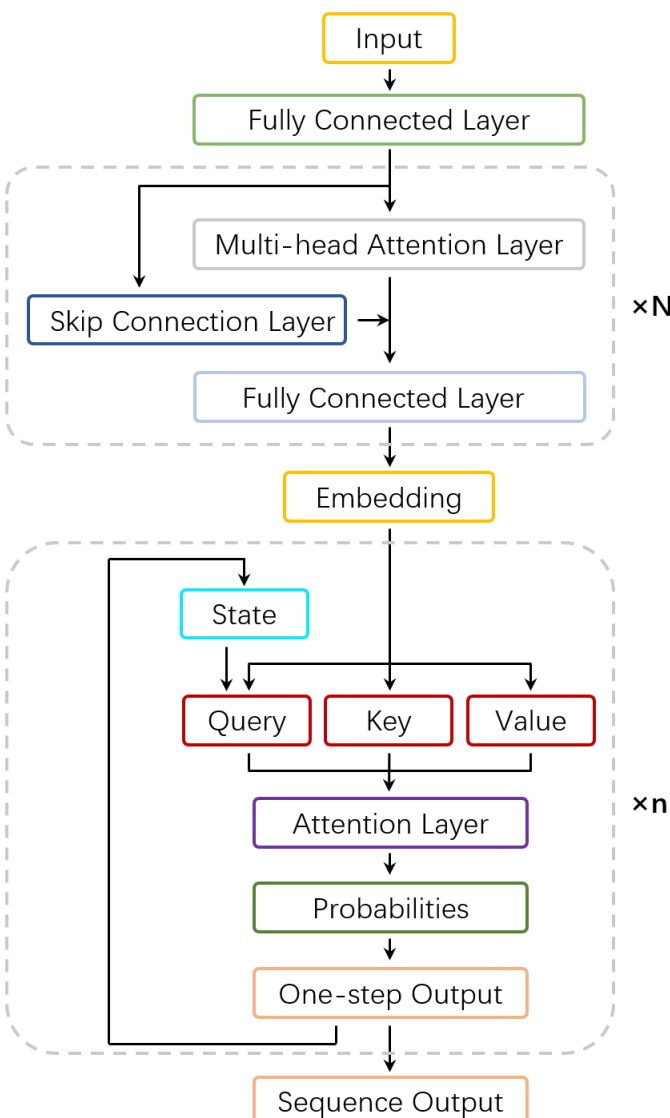

**Figure 1.** The structure of the proposed model. At first, the graph embeddings are calculated by the full-graph encoder with *N* layers, which include multi-head attention layers, skip connection layers, and fully connected layers. Then, a one-step output can be calculated by the two-way selection decoder. Finally, the sequence can be output after the decoder has run for *n* times.

### 3.1. Full-Graph Encoder

This part studies the structure of the full-graph encoder. The full-graph encoder constructs the graph structure to transform real-world problems into graph problems. It extracts features from individual nodes and outputs the node embeddings.

The full-graph encoder builds a graph and transforms the information of each point in real life (depots and tasks) into a feature vector that can be recognized by the machine. Then, the encoder calculates the correlation among nodes using the multi-head attention mechanism and outputs the node embeddings and graph embedding. The "full-graph" attribute of the encoder is reflected in the unified processing of all depots and all tasks. The structure of the encoder is shown in Figure 2.

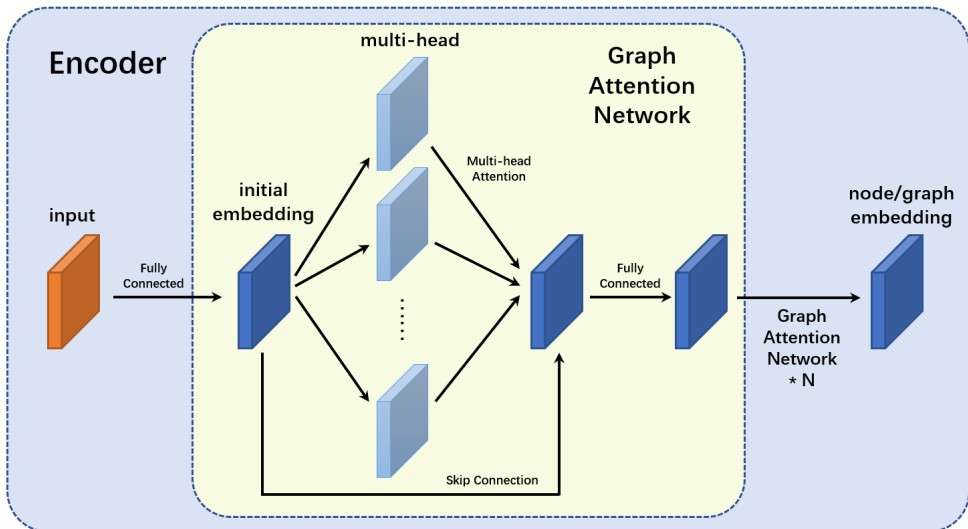

**Figure 2.** The structure of the proposed full-graph encoder. At first, the encoder takes as input the coordinates of task points and agent depots and then moves to the next step, which calculates the initial embeddings using a fully connected layer. Then, the embeddings of the hidden layer are calculated by the graph attention network, which is the combination of a multi-head layer, a skip connection layer, and a fully connected layer. Finally, the node embeddings and the graph embedding can be calculated after $N$ graph attention networks.

Firstly, the full-graph encoder constructs an undirected fully connected graph. The points in the real-world problem are mapped to the graph nodes one by one. In this multi-depot problem, there are $m$ depots $\{A_1, \ldots, A_m\}$ and $n$ tasks $\{T_1, \ldots, T_n\}$, so a graph with $m + n$ nodes can be built. In this graph, each node will be connected to all other nodes and also to itself, and all connections are undirected.

Secondly, the full-graph encoder generates a $\gamma$ dimension vector ($\gamma = 128$ in this paper) for each node using the position information. The input comes from the real problem, including the coordinates of all depots $\{\mathcal{C}_{A_1}, \ldots, \mathcal{C}_{A_m}\}$ and all tasks $\{\mathcal{C}_{T_1}, \ldots, \mathcal{C}_{T_n}\}$. Then, a fully connected neural network is used to transform the coordinate vectors into $\gamma$-dimensional feature vectors, completing initialization for the subsequent multi-head attention network. The initial node embedding $E^{(0)}$ is calculated after passing through the fully connected layer. In this paper, the uppercase letter $E$ is used to indicate the embedding of all nodes in the graph, and the lowercase letter $e$ is used to indicate the embedding of one node. $E^{(0)}$ includes all $m + n$ node embeddings $e_i^{(0)}$, where $i = 1, \ldots, m + n$.

Finally, the node embeddings and graph embedding are calculated by the multi-head attention mechanism. In the complex multi-depot problem, shallow embeddings cannot effectively represent the features of nodes (the network depth is not enough). Therefore, this paper adopts improved graph attention networks to carry out calculations for deeper embedding. Each graph attention network contains a multi-head attention layer [18], a fully connected layer, and a skip connection layer [25]. The graph attention network is shown in Figures 1 and 2. The multi-head attention layer is the core structure of the graph attention network; it aims to obtain the features of each node and the correlation among multiple nodes. The fully connected layer ensures that the input and output have the same dimension of t, allowing the graph attention networks to stack with each other. The skip connection layer retains the useful features from the previous layers to prevent them becoming invisible in subsequent calculations. Increasing network depth is crucial in deep neural networks, so the skip connection layer is also crucial to avoid the problem of gradient vanishing and exploding. The number of the stacked graph attention networks $N$ should be selected moderately ($N = 3$ in this paper).

The multi-head attention layer is the core of the graph attention network, and its structure is shown in Figure 3. It can update from the current embedding $E^{(l)}$ to the

next layer $E^{(l+1)}$. Specifically, for the embedding $e_i^{(l)}$ of the $l$-th layer, the $r$-th single-head attention layer can calculate the query $q_{i,r}^{(l)}$, key $k_{i,r}^{(l)}$ and value $v_{i,r}^{(l)}$ according to the parameters $w_r^{q_i(l)}$, $w_r^{k_i(l)}$, and $w_r^{v_i(l)}$:

$$
\begin{aligned}
q_{i,r}^{(l)} &= e_i^{(l)} \cdot w_r^{q_i(l)} \ , \ r = 1, \ldots, h \\
k_{i,r}^{(l)} &= e_i^{(l)} \cdot w_r^{k_i(l)} \ , \ r = 1, \ldots, h \\
v_{i,r}^{(l)} &= e_i^{(l)} \cdot w_r^{v_i(l)} \ , \ r = 1, \ldots, h
\end{aligned}
\tag{7}
$$

where $h$ represents the number of heads in the multi-head attention layer.

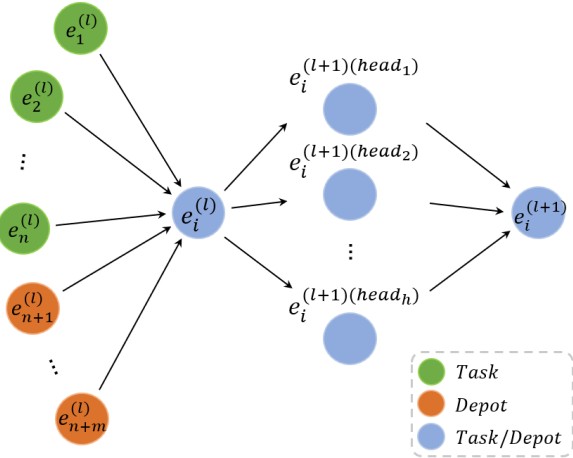

**Figure 3.** The details of the multi-head attention sublayer. Firstly, the weight vector of each node is calculated by the query of the node itself and the keys of all nodes. Secondly, the result of a single-head attention layer is obtained by weighted averaging the values of all nodes using the weight vector. Finally, the output of the multi-head attention layer can be obtained by stacking all single-head results.

After obtaining the query $q_{i,r}^{(l)}$, key $k_{i,r}^{(l)}$, and value $v_{i,r}^{(l)}$ of the $r$-th head, the embedding $e_{i,r}^{(l+1)}$ of the next layer can be calculated by:

$$
e_{i,r}^{(l+1)} = \sum_{s=1}^{n} softmax\left(\frac{< q_{i,r}^{(l)}, k_{s,r}^{(l)} >}{\sqrt{d_k}}\right) v_{s,r}^{(l)}
\tag{8}
$$

In this formula, the query $q_{i,r}^{(l)}$ can give weights to other nodes by calculating the matching degree, which is equal to the cosine value of the angle between these two vectors $q_{i,r}^{(l)}$ and $k_{s,r}^{(l)}$. Then, the softmax function is used to calculate the weight from the cosine value. The greater the cosine value, the greater the weight given. The adjustment factor $\sqrt{d_k}$ is used to prevent the result of softmax being too close to 0 or 1. Finally, the values $v_{s,r}^{(l)}$ of all nodes are linearly combined to calculate the node embedding $e_{i,r}^{(l+1)}$.

The multi-head attention layer repeats the above single-head attention layer several times, and outputs $\hat{e}_i^{(l+1)}$ by splicing all heads $e_{i,r}^{(l+1)}$:

$$
\hat{e}_i^{(l+1)} = (e_{i,1}^{(l+1)}, \ldots, e_{i,h}^{(l+1)})
\tag{9}
$$

Afterwards, the result of the graph attention network $e_i^{(l+1)}$ can be calculated by $\hat{e}_i^{(l+1)}$ after passing through a fully connected layer and a skip connection layer:

$$e_i^{(l+1)} = w_i^{(l+1)}(\hat{e}_i^{(l+1)}) + b_i^{(l+1)} + e_i^{(l)} \tag{10}$$

It is worth noting that the dimension of each head's embedding $e_{i,r}^{(l+1)}$ is the quotient of the embedding dimension $\gamma$ and the number of heads $h$. The dimension of the vector must be an integer, so $\gamma$ must be divisible by $h$. In addition, the parameters $w_r^{q_i(l)}$, $w_r^{k_i(l)}$, and $w_r^{v_i(l)}$ of each head are different to extract different features.

### 3.2. Two-Way Selection Decoder

This part studies the structure of the two-way selection decoder. The two-way selection decoder reasonably allocates tasks to multiple agents and arranges the task execution orders of individual agents.

The pointer-network-based decoder is a sequence generator, but the classical pointer network can only output one pointer, which cannot achieve the goal of selecting an agent for a task in a multi-depot problem. Therefore, this paper proposes a two-way selection pointer network that can output two pointers at the same time, allocating tasks reasonably to agents while considering the task execution order.

The routes of all agents (sequences of all tasks) can be obtained when the two-way selection decoder runs multiple times. Each run of the two-way selection decoder obtains the best single step in the current state, and updates the state information every step. All tasks can be allocated after the decoder repeatedly runs. The process of the decoder outputting multiple routes is shown in Figure 4.

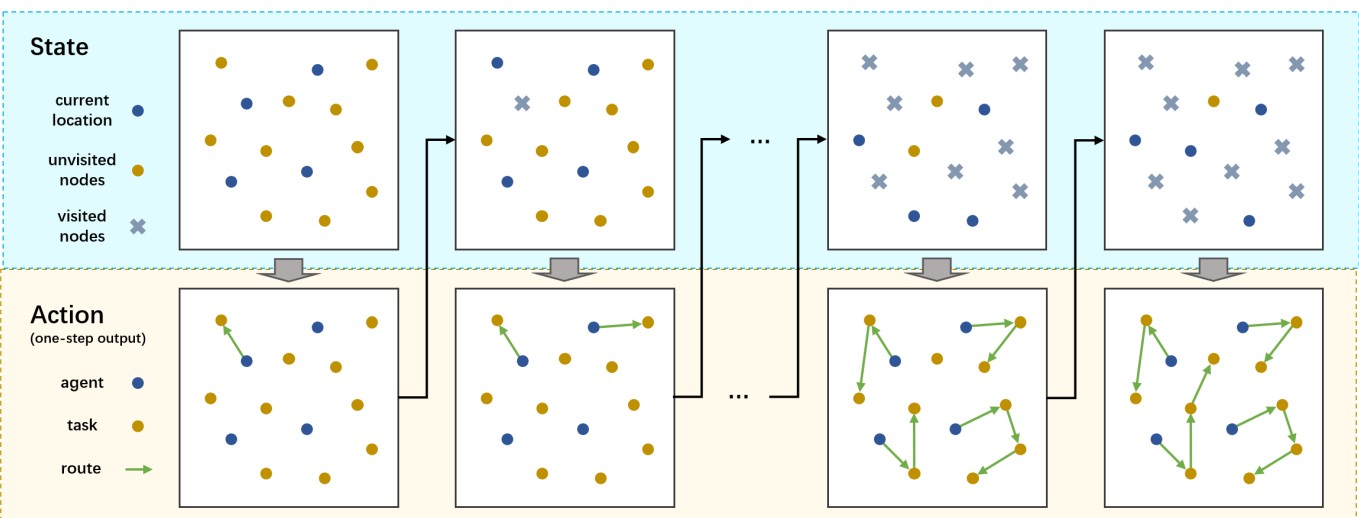

**Figure 4.** The process of obtaining the final solution from the problem example. Each action represents an agent executing a task, then each state updates the position of the agent and moves the executed task out of the candidate task set. The final strategy can be obtained by repeating actions and updating states continuously until all tasks are completed.

The input of the decoder includes a static part and a dynamic part. The node embeddings obtained in the full-graph encoder are used as the static input of the decoder. The dynamic input includes all information about the current state, which is an important factor affecting the final results. The current location of the agents will obviously affect the node selection in the multi-depot problem, and the "visited state" of all nodes will influence the results as well. To be specific, the "visited state" means that only the tasks that have not been visited can become candidates, while the visited tasks cannot be selected again. So,

the dynamic input consists of the current locations of agents and the tasks that have not been visited.

The outputs of the decoder are the routes of multiple agents, corresponding to several sequences of all tasks, and its set $\pi(s)$ can be represented as:

$$\pi(s) = \{\pi_1(s), \ldots, \pi_m(s)\} \tag{11}$$

where $\pi(s)$ is the set of all task execution routes of $m$ agents, which is the same as the allocation plan in Section 2.

The structure of the two-way selection decoder is shown in Figure 5.

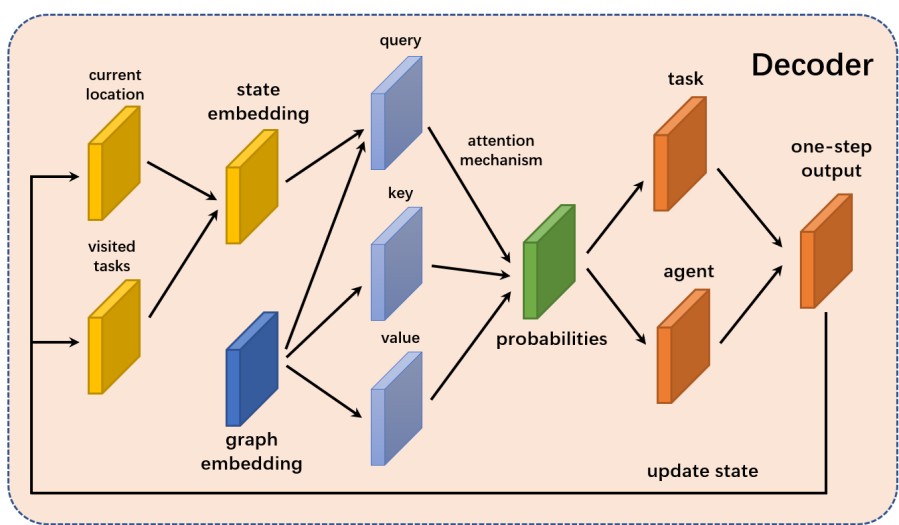

**Figure 5.** The structure of the two-way selection decoder. Firstly, the decoder calculates the state embedding with the current location of agents and the set of candidate tasks (visited tasks). Secondly, the attention mechanism is used to obtain the probability of each agent executing each task. Thirdly, an agent and a task are chosen as a one-step output. Finally, the state of the current location of agents and the set of candidate tasks can be updated.

In fact, the output of the two-way selection pointer network is the probability distribution $p(A_j, T_i|t)$ of selecting each node in the current state, where $A_j \in \{A_0, \ldots, A_m\}$ and $T_i \in \{T_0, \ldots, T_n\}$ represent the selected agent and task. The calculation of the probability distribution is an important process of the network, and its core principle is similar to the weight calculation process of the attention mechanism. When calculating the probability, the query, key, and value vectors are crucial. The key and value are produced from the static node embeddings, while the query is dynamic, which directly affects the quality of the results. A critical part of the network is whether the query can fully obtain the state information.

The core concept of the two-way selection pointer network is to design two queries, one for an agent and the other for a task. The query $q_j^{(A)}$ for the agent is calculated by the current location, completing the selection of a task that is suitable for the agent. On the contrary, the query $q^{(T)}$ for the task comes from its node embedding and the "visited state", completing the selection of an agent that is suitable for the task. Then, the comprehensive probability is calculated based on the results of the two selections, which better reflects the matching situation between the agent and the task. The detailed calculation process of $q_j^{(A)}$ and $q_i^{(T)}$ are as follows:

$$q_j^{(A)} = FC(e_j^{(t)})$$
$$q_i^{(T)} = FC(e_i, \overline{e(u)}) \tag{12}$$

where $e_j^{(t)}$ represents the embedding of the task of the $j$-th agent located at moment $t$. $\overline{e(u)}$ represents the average embedding of all tasks that have not been visited. *FC* means a fully connected layer whose output dimension is the embedding dimension $\gamma$.

After obtaining $q_j^{(A)}$ and $q_i^{(T)}$, the relationship $\mathcal{R}_{i,j}$ between agent $j$ and task $i$ can be calculated by the two-way selection rules:

$$\mathcal{R}_{i,j} = \frac{< q_j^{(A)}, k_i^{(T)} > \cdot < q_i^{(T)}, k_j^{(A)} >}{d_k} \tag{13}$$

Subsequently, in order to satisfy the constraint of non-repeated task visitation, it is necessary to add a mask function. The purpose of the mask function is to ensure that the probability of all visited tasks is equal to 0. The mask function searches the relationships $\mathcal{R}_{i,j}$ of all visited tasks and changes them to negative infinity, so that the probabilities can be equal to 0 after the softmax function. The calculation process of the mask function is as follows:

$$Mask(\mathcal{R}_{i,j}) = \begin{cases} -\infty \ , \ visited \\ \mathcal{R}_{i,j} \ , \ others \end{cases} \tag{14}$$

Then, the probability $p(A_j, T_i|t)$ of selecting agent $j$ and task $i$ can be calculated by the softmax function:

$$p(A_j, T_i|t) = softmax(Mask(\mathcal{R}_{i,j})) \tag{15}$$

After calculating the probability distribution, the decoder will select one agent and one task according to the probability distribution to obtain a single-step output. The algorithm for selecting the single-step output varies in different circumstances, and it comes from the following three algorithms: greedy search, beam search, and sampling selection.

When training, the sampling selection algorithm should be chosen. It will randomly select a node according to the probability distribution $p(A_j, T_i|t)$. This algorithm can hardly obtain a good solution, but is more suitable for gradient calculation.

When testing, the greedy search or beam search algorithm should be chosen. The greedy search algorithm selects the highest probability every time. The greedy search algorithm has a very high computational efficiency, but ultimately the performance of the feasible solution is not always the best. The beam search algorithm selects several high probabilities as candidates every time, and finally selects the best plan from all candidates, resulting in better performance but lower computational efficiency.

### 3.3. Reinforcement with Baseline

The policy gradient deep reinforcement learning algorithm with baseline [21] is used to train the model due to the difficulty of collecting labels. In the policy gradient algorithm, the probability of the action is determined by the state, where the action is the behavior of the node selection. A loss function is needed for the policy gradient algorithm, so that it can use a gradient descent method to minimize the loss and to determine a model. Based on the model parameters, $\mathcal{W}$, $p(\pi|s, \mathcal{W})$ is the probability distribution of all feasible solutions under the problem instance $s$, then the loss function can be defined as:

$$\mathcal{L}(\theta|s) = E_{p_\theta(\pi|s)}[\mathcal{J}(\pi|(s))] \tag{16}$$

$$\mathcal{J}(\pi|(s)) = \sum_{j=1}^{m} \mathcal{L}(\pi_j(s)) \tag{17}$$

where $\mathcal{L}(\theta|s)$ indicates the loss function, and $\mathcal{J}(\pi|(s))$ indicates the cost (the length of all routes). Then, the gradient of the loss function is:

$$\nabla \mathcal{L}(s, \theta) = E_{p(\pi|s, \theta)}[\mathcal{J}(\pi|(s))\nabla \ln p(\pi|s, \theta)] \tag{18}$$

In the calculation of the gradient, the addition of baseline $b$, which is independent of $\pi$, does not affect the result, because $\forall b$ satisfies:

$$E_{p(\pi|s,\theta)}[b\nabla \ln p(\pi|s,\theta)] = b \cdot E_{p(\pi|s,\theta)}[\nabla \ln p(\pi|s,\theta)] = b \cdot \sum_{\pi} p(\pi|s,\theta) \cdot \nabla \ln p(\pi|s,\theta)$$

$$= b \cdot \sum_{\pi} p(\pi|s,\theta) \cdot \frac{\partial \ln p(\pi|s,\theta)}{\partial\theta} = b \cdot \sum_{\pi} p(\pi|s,\theta) \cdot \frac{1}{p(\pi|s,\theta)} \cdot \frac{\partial p(\pi|s,\theta)}{\partial\theta} \tag{19}$$

$$= b \cdot \sum_{\pi} \frac{\partial p(\pi|s,\theta)}{\partial\theta} = b \cdot \frac{\partial \sum_{\pi} p(\pi|s,\theta)}{\partial\theta} = b \cdot \frac{\partial 1}{\partial\theta} = 0$$

Therefore, the gradient of the loss for the problem instance $s$ can be expressed as:

$$\nabla\mathscr{L}(s,\theta) = E_{p(\pi|s,\theta)}[\mathcal{J}(\pi|(s))\nabla \ln p(\pi|s,\theta)]$$
$$= E_{p(\pi|s,\theta)}[(\mathcal{J}(\pi|(s)) - b)\nabla \ln p(\pi|s,\theta)] \tag{20}$$

So $(\mathcal{J}(\pi|(s)) - b)$ can be used instead of $\mathcal{J}(\pi|(s))$ in the policy gradient as long as $b$ is not related to action $\pi$. The addition of $b$ will not affect the correctness of the policy gradient itself. More importantly, the policy gradient algorithm in deep reinforcement learning is performed from Monte Carlo approximation, which is a kind of sampling method. When the baseline $b$ is close to $\mathcal{J}(\pi|(s))$, the variance can be reduced so that the convergence of the neural network can speed up [26].

The baseline in the model is as follows:

$$\begin{cases} b(s) = \mathcal{J}(\pi|(s)) \,, \; first \; iteration \\ b(s) \leftarrow \beta b(s) + (1-\beta)\mathcal{J}(\pi|(s)) \,, \; others \end{cases} \tag{21}$$

where $b(s)$ is the cost of the baseline policy, and $\beta$ is an inertia parameter.

## 4. Experiments and Results

The experiments in this paper are conducted through random positions of task points and agent depots.

This section is organized as follows. Section 4.1 gives the parameter settings and the training resources in this paper. Section 4.2 describes the comparative experiment and the numerical results. Section 4.3 shows some figures of the experiment. Section 4.4 provides the analysis of the model based on the experimental results.

### 4.1. Parameter Settings and Training Details

In most deep learning models, the construction of training and validation sets is crucial. The model can only use training data for training and cannot use samples in the validation set. In this paper, a set of validation samples is generated before training, with the number of agents and task points in the validation set unchanged. The positions of agent depots and task points in each sample are randomly generated in a $1 \times 1$ map. The generation of the training set is the same as the validation set, using a fixed quantity and random location. However, due to the large number of samples in the training set, it is necessary to randomly divide it into multiple batches for training before each epoch training. Then, the model is validated one time after completing each epoch. Specifically, the policy gradient algorithm calculates the baseline required for validation, then the loss between the model and the baseline on the validation set is calculated and finally uses the Adam optimization algorithm to update the weights of the neural network. The final result of each training is selected from the model that performs best in the validation set. Some model parameters and training resources are shown in Table 1 for reference.

**Table 1.** Some model parameters.

| Parameter | Value |
|---|---|
| Embedding dimension | 128 |
| Hidden layer dimension | 128 |
| Attention layer number | 3 |
| $\beta$ | 0.8 |
| Batch size | 256 |
| Epoch size | 12,800 |
| Epoch number | 100 |
| Validation size | 1000 |
| Learning rate | 0.0002 |

In the above table, the embedding dimension is for the graph encoder, and the hidden layer dimension is for both the encoder and decoder. The attention layer number refers to the number of multi-head attention layers in the encoder. The inertia parameter $\beta$ is defined in Equation (21). The batch size is set to 256, and a total of 100 epochs are trained with each epoch containing 12,800 batches. The validation size means the number of instances in the validation set. The learning rate is set to 0.0002.

In addition, the proposed graph attention pointer network was trained on 2 GPUs "NVIDIA Tesla V100 sxm30 32 GB" for about 6 h.

### 4.2. Comparative Experiment

The Gurobi and LKH3 methods, which are two commonly used solvers, are selected as comparative experiments. The Gurobi method is one of the most widely used mathematical programming optimizers; it is often used to solve integer programming problems. However, due to limited equipment resources, Gurobi cannot be used to handle larger-scale problems. The LKH3 is an algorithm that optimizes the solution by constantly exchanging edges; it is an excellent solver for TSP and VRP at present. For the LKH3 algorithm, the higher the number of exchanged edges, the better the quality of the solution. In this paper, the solution of the LKH3 algorithm is defined as a solution calculated within an acceptable time. Meanwhile, for the comparison purpose, the particle swarm optimization (PSO) and the genetic algorithm (GA) are also designed to solve this problem. PSO is a well-known heuristic algorithm which is able to deal with large-scale problems. GA can hardly deal with large-scale problems due to the huge memory resources required during genetic computing, so GA is not available when there are more than 10 agents. As for deep learning methods, there are few similar deep learning methods for reference due to the fact that the method in this paper is an extension of the attention model [4]. The simulation results of the five methods are shown in Table 2.

**Table 2.** The results of different models.

| Map Scale | Our Model | | Gurobi | | LKH3 | | PSO | | GA | |
|---|---|---|---|---|---|---|---|---|---|---|
| Agent/Task | Length | Time | Length | Time | Length | Time | Length | Time | Length | Time |
| 5/20 | 4.67 | <0.1 s | 4.17 | 3 m | 4.35 | 9 m | 4.56 | 23 s | 4.51 | 37 s |
| 5/50 | 7.23 | <0.1 s | - | - | 7.09 | 23 m | 7.89 | 56 s | 8.12 | 2 m |
| 10/50 | 6.91 | <0.1 s | - | - | 6.83 | 2 h | 8.10 | 5 m | - | - |
| 10/100 | 9.89 | <0.1 s | - | - | 9.83 | 13 h | 11.39 | 12 m | - | - |

Overall, the most obvious advantage of the method proposed in this paper lies in the computational speed, which can be faster by over 100 times compared to all optimization methods and heuristic algorithms mentioned in this paper (such fast computing speed is based on a long training period). In terms of performance, the method proposed in this paper is not as good as the optimization methods but performs better than heuristic algorithms in large-scale problems.

Specifically, the performance of the deep learning method proposed in this paper varies among problems with different scales. At the amount of five agents with 20 task points, the performance of our method lags behind all other methods. Moreover, in this scale of problems, the optimization methods and heuristic algorithms also have fast computational speed, so the advantage of a fast speed does not seem to be important. At the amount of five agents with 50 task points, the performance of our method lags behind LKH3 but leads PSO and GA. At the amount of 10 agents with 50 task points and 100 task points, the performance of our method lags behind LKH3 but significantly leads PSO and GA, and in such large-scale problems, the advantage of the fast speed in our method is even more important (not all problems can accept taking 13 h to find a solution).

### 4.3. Figures

The performance and characteristics of this method can be further explored from different simulation figures.

The convergence curve of the loss function is shown in Figure 6. In the training process of 100 epochs, the loss function exhibits a decaying trend along with the number of epochs and is very close to the minimum value after the 80th epoch.

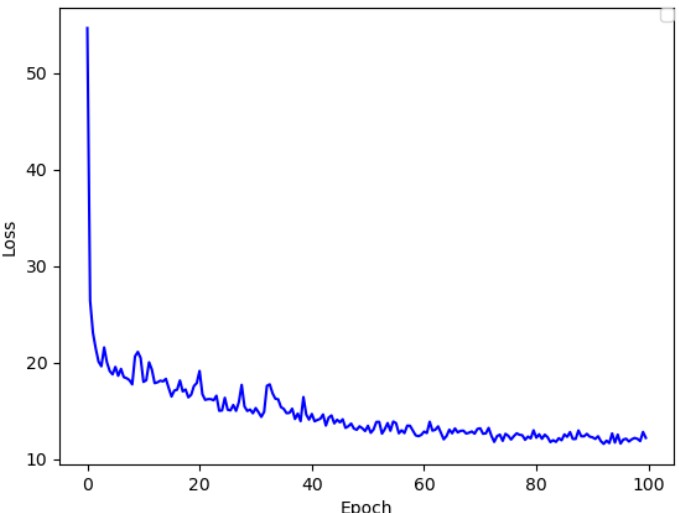

**Figure 6.** The convergence curve of the loss function.

The visualization graphs can demonstrate the quality of the task allocation results, and can also be used to analyze the advantages and drawbacks of the allocation results. The visualization graphs of the multi-depot problem with 100 tasks and 10 agents are shown in Figure 7.

In Figure 7, it can be found that the points distributed at the edge of the map are selected much more preferentially than the dense points in the center. As a result, the sequence is always generated from the edge to the center.

Additionally, the two models of Figure 7a,b yield completely different strategies for the task allocation. The model of (a) tends to select neighbor points with a closer distance, while the model of (b) gives priority to placing tasks of one path in the same area. The reason for this phenomenon might be that the two models learned different features during the training process. One model learns more about the distribution of nodes, and the other learns more about the spatial distance between nodes.

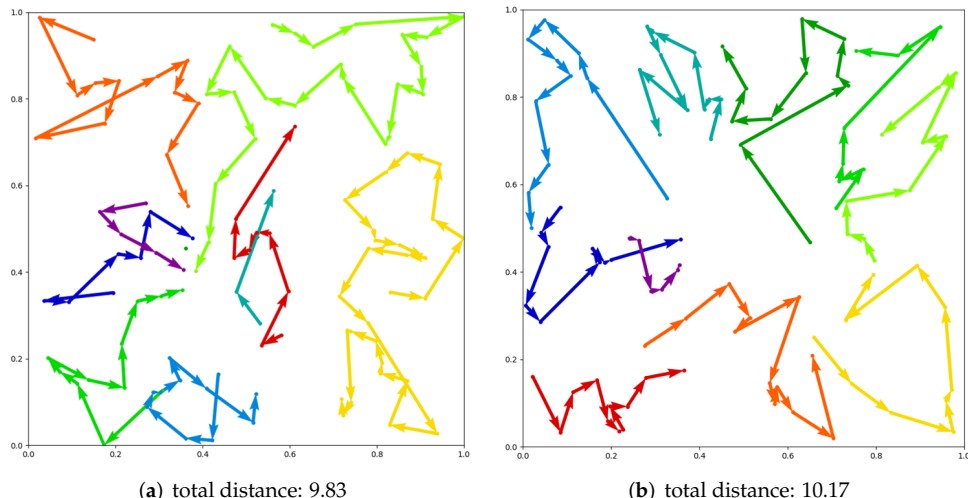

(**a**) total distance: 9.83          (**b**) total distance: 10.17

**Figure 7.** The visualization results of the simulation of 100 tasks for 10 agents. The lines of different colors are used to represent the routes of different agents, and the agent will be represented by a point if it is not assigned to any task.

*4.4. Analysis*

According to the simulation results in Section 4.2, the deep learning method proposed in this paper has many advantages. Firstly, the deep learning method performs well in large-scale problems and can be applied to some complex problems. Secondly, the deep learning method has great computational efficiency that can handle some dynamic problems which require high computing speed. Finally, the deep learning method can collect lots of labels in a short time, which is helpful for many practical applications and supervised learning models.

On the contrary, there are several drawbacks of the proposed deep learning method. Firstly, a new model needs to be built when the problem description changes. Secondly, the deep learning method needs a long training period, even if the problem is very simple. Thirdly, the deep learning method cannot yield an optimal solution, so it is not as reliable as classic solvers for some problems with high performance requirements. Finally, the deep learning method yields a black-box model which may not be interpretable, so the strategy may not be reasonable.

In the research of task allocation, various methods have their advantages and limitations. All methods can find suitable application scenarios based on their own characteristics and will not be completely replaced by other methods. Deep learning methods change the solving mode of task allocation problems from the all-points output to point-by-point output. This characteristic of deep learning methods makes the computational complexity only dependent on the network itself and almost unchanged with a change in the problem size. Therefore, deep learning methods can maintain highly efficient computation and good performance even when the problem size is large. Compared to heuristic algorithms, deep learning methods lack flexibility. When the problem becomes complex or the objective changes, deep learning methods not only need to redesign all the networks but also spend a lot of time adjusting parameters to complete training, while heuristic algorithms only need to change the objective function and constraint conditions. Compared to optimization methods, the performance of deep learning methods is not stable enough. When the tolerance for poor performance is low, deep learning methods are often inadequate. Deep learning methods may not perform well in certain examples, but optimization methods can always obtain excellent solutions.

In summary, deep learning methods are better when the problem is large in scale or the problem requires high computational speed; heuristic algorithms are better when the



problem is complex enough or there are changes in objectives; optimization methods are better when there is a high demand for optimality.

## 5. Conclusions

In this paper, a multi-agent task allocation method with multiple depots has been proposed using the graph attention pointer network. Specifically, a full-graph encoder with multi-head attention mechanism and a two-way selection decoder based on a pointer network have been built to form an encoder–decoder architecture. Simulation results show that the proposed task allocation method outperforms the traditional solvers and heuristic algorithms in terms of computational speed and training resource consumption.

Our future work will further improve the network. A dynamic graph embedding structure will be built, and the multi-head attention mechanism will be integrated into the calculation of the state embedding in the decoder in order to improve the quality of the state features. Moreover, the decoder structure will be modified in order to be applied to more complex problems. In addition, the mode of training and the generation of data will also be changed to reduce the time for training.

**Author Contributions:** Conceptualization, W.S. and C.Y.; methodology, W.S. and C.Y.; software, W.S.; validation, W.S.; formal analysis, W.S. and C.Y.; investigation, W.S. and C.Y.; resources, C.Y.; data curation, W.S.; writing—original draft preparation, W.S.; writing—review and editing, C.Y.; visualization, W.S.; supervision, W.S. and C.Y.; project administration, C.Y.; funding acquisition, C.Y. All authors have read and agreed to the published version of the manuscript.

**Funding:** This work was supported by the National Key Research and Development Project under grant 2020YFC1512503, and the National Natural Science Foundation of China (grant no. 61991414, 62088101).

**Data Availability Statement:** In this paper, we use randomly generated data.

**Conflicts of Interest:** The authors declare no conflict of interest.

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
