# Peer review of "Multi-Agent Task Allocation with Multiple Depots Using Graph Attention Pointer Network"

_electronics, doi:10.3390/electronics12163378_

Round 1

Reviewer 1 Report

Title: Multi-agent Task Allocation with Multi-depot Using Graph Attention Pointer Network

Authors: Wen Shi* , Chengpu Yu

The article engages with the advancement of a Graph Attention Pointer Network for multi-agent task allocation with multiple depots. The authors stress that multi-agent collaboration, particularly when there are multiple hubs of operation, is a complex problem that current heuristic algorithms struggle to efficiently address. These inefficiencies can lead to suboptimal task distribution, reduced system performance, and subsequently lower productivity and resource utilization. In this context, the authors present a novel Graph Attention Pointer Network designed to optimize multi-agent task allocation. The cornerstone of this network is the application of a multi-head attention mechanism for comprehensive node feature extraction. In addition, the paper introduces a pointer network which incorporates parallel two-way selection and output, designed to boost the overall performance of multi-agent cooperation and increase the efficiency of task allocation. Through experimental results, the authors demonstrate the superiority of their graph attention pointer network over traditional heuristic algorithms, affirming its potential in improving multi-agent task allocation. This further substantiates the need for advanced computational models in addressing complex collaborative tasks in multi-agent systems.

The overall structure and presentation of the paper is good. There are, however, some issues that need to be addressed in order to improve the quality of the paper:

-------------------

1) The caption of all figures is not descriptive enough. People with eyesight problems who are not able to view the images and use text to speech software should be able receive the same information via the caption of the image. For example, the caption of Figure 2 should be something like: "At first, the algorithm takes as input XXX and then moves to the next step, which calculates YYY. Then, ZZZ is performed... Finally, the output of the algorithm includes...". It is not necessary to be that detailed, but I woule like to see something morethan "The structure of the proposed encoder".

2) The experimental setup and choice of algorithms to compare need more details. Why did the authors choose Gurobi and LKH3 for comparisons? Does the proposed algorithm outperform traditional models in every scenario? Are there any scenarios in which the proposed algorithm performs worse? Please give more details.

3) Even though the authors talk about the drawbacks of their implementation (Line 359), I would like to see more details. What do these drawbacks mean in real world applications? Do the advantages outweigh  the disadvantages?

4) The authors should include a paragraph that explains future possible improvements of the proposed system. How could the algorithm for example be further improved in terms of complexity, speed and effectiveness? Please include a paragraph in the Conclusions section.

Reviewer 2 Report

This paper proposes a novel approach to solving the multi-agent task allocation problem, which is crucial for investigating multi-agent collaboration. The paper presents a graph attention pointer network that employs a multi-head attention mechanism for feature extraction and a pointer network with parallel two-way selection and parallel output to improve the performance of multi-agent cooperation and task allocation efficiency. The paper also includes a detailed experimental evaluation of the proposed approach and compares it with other state-of-the-art methods.

Pros:

1. The paper introduces a novel Graph Attention Pointer Network to tackle the multi-agent task allocation problem with multiple depots. This approach, which combines a multi-head attention mechanism and a pointer network, is innovative and practical, offering a new perspective on solving complex task allocation problems.

2. The paper presents a thorough theoretical analysis of the multi-agent task allocation problem, including the formulation of the problem and the design of the proposed solution. This in-depth analysis contributes to a better understanding of the problem and the effectiveness of the proposed solution.

3. The authors provide empirical evidence showing that their proposed algorithm outperforms traditional heuristic algorithms in terms of effectiveness and efficiency. This validates their approach and emphasizes the practical value of their research.

Cons:

1. While the paper introduces an innovative approach to the multi-agent task allocation problem, it lacks sufficient details on the implementation of the proposed Graph Attention Pointer Network. More specifics on the architecture, parameters, and training process would be beneficial for readers aiming to reproduce or build upon this study.

2. The paper mentions that the proposed method outperforms traditional heuristic algorithms, but it does not provide a detailed comparison with widely-known baselines in the field. Including such comparisons would strengthen the paper's claims and provide a clearer context for the proposed method's performance.

3. Some parts of the paper could benefit from clearer exposition. For example, the description of the two-way selection network and the parallel pointer network could be more detailed to aid reader comprehension. Providing more illustrative examples or visual aids might also improve the paper's readability.

4. The paper assumes a certain level of background knowledge in multi-agent systems, task allocation problems, and deep learning methodologies like the attention mechanism and pointer networks. While this is common in academic papers, providing a brief overview or explanation of these concepts could make the paper more accessible to readers who are new to these topics.

5. The paper could provide a more detailed discussion on the limitations of the proposed method. Every method has its limitations and understanding these can provide valuable insights for future research. This would also demonstrate the authors' thorough understanding of their work.

This paper presents an innovative approach to the multi-agent task allocation problem with multiple depots. The authors' proposed Graph Attention Pointer Network shows promising results, outperforming traditional heuristic algorithms. However, the paper could benefit from more detailed implementation information, extensive evaluation studies, and clearer exposition. Despite these areas for improvement, the paper is a valuable contribution to the field.
